# Patient-Reported Financial Distress in Cancer: A Systematic Review of Risk Factors in Universal Healthcare Systems

**DOI:** 10.3390/cancers13195015

**Published:** 2021-10-07

**Authors:** Sophie Pauge, Bastian Surmann, Katja Mehlis, Andrea Zueger, Luise Richter, Natalja Menold, Wolfgang Greiner, Eva C. Winkler

**Affiliations:** 1Department for Health Economics and Health Care Management, School of Public Health, Bielefeld University, 33615 Bielefeld, Germany; bastian.surmann@uni-bielefeld.de (B.S.); wolfgang.greiner@uni-bielefeld.de (W.G.); 2Section of Translational Medical Ethics, Department of Medical Oncology, National Center for Tumor Diseases (NCT), Heidelberg University Hospital, 69120 Heidelberg, Germany; katja.mehlis@med.uni-heidelberg.de (K.M.); andrea.zueger@med.uni-heidelberg.de (A.Z.); eva.winkler@med.uni-heidelberg.de (E.C.W.); 3Methods in Empirical Social Research, Institute of Sociology, Faculty of Arts, Humanities and Social Science, Dresden University, 01062 Dresden, Germany; luise.richter4@tu-dresden.de (L.R.); natalja.menold@tu-dresden.de (N.M.)

**Keywords:** cancer, financial toxicity, financial distress, universal healthcare coverage, risk factors

## Abstract

**Simple Summary:**

A comprehensive understanding of risk factors associated with experiencing subjective financial distress is needed to inform the development of valid instruments and effective interventions to tackle financial toxicity. Several studies from the US indicate a strong correlation of the US-healthcare system’s systematic organisation and its particular socioeconomic risk factors for cancer patients experiencing financial toxicity. It is assumed that risk factors differ in other high-income countries due to the different structure of universal healthcare coverage. As an exhaustive analysis for other countries is lacking, this review aims to identify risk factors for subjective financial distress in universal healthcare systems.

**Abstract:**

Financial toxicity is a side effect of cancer that results from the perceived financial distress an individual may experience in the course of the disease. The purpose of this paper is to analyse underlying factors related to subjective financial distress in high-income countries with universal healthcare coverage. A systematic literature review was conducted to identify qualitative and quantitative studies of cancer patient-reported subjective financial distress by performing a search in the databases of PubMed, PsycINFO and CINAHL up to December 2020. A qualitative synthesis was performed linking the time-dependent occurrence of risk factors to derived categories of risk factors. Out of 4321 identified records, 30 quantitative and 16 qualitative studies were eligible. Classification of risk factors resulted in eight categories with a total of 34 subcategories. Subjective financial distress is primarily determined by pre-diagnosis sociodemographic- factors as well as financial and work factors that might change during the course of the disease. The design of healthcare and social security systems shapes the country-specific degree of subjective financial distress. Further research should focus on evolving multidisciplinary intervention schemes and multidimensional instruments for subjective financial distress to account for identified risk factors in universal healthcare systems more precisely.

## 1. Introduction

In recent years, the rising understanding of cancer as a chronic disease with lifelong individual consequences has shifted the awareness of potential side effects beyond clinical outcomes [1]. Due to direct implications of the disease, such as direct (non-)medical costs and the ability to work, it is now acknowledged that cancer patients may experience long lasting financial consequences [2]. The individual patients may, however, respond differently to objective financial burden, which is defined as the direct and indirect costs of the disease. Patients’ reactions are, for example, influenced by personal pre-conditions and coping patterns determining the degree of subjectively experienced financial distress [3]. Following the framework introduced by Witte et al. [4], the outcome of subjective financial distress can be described as “financial toxicity”. More recently, Lueckmann et al. [5] expanded the understanding of subjective financial distress, where distress is not only recognised as a direct consequence of objective financial burden but also includes the expectation of experiencing the burden. Thus, the individual assessment of subjective financial distress has a salient role within the concept of financial toxicity and is therefore the starting point for evaluating financial toxicity. Studies showed that experiencing high degrees of subjective financial distress, hence suffering from financial toxicity, not only deteriorates the quality of life significantly [6] but might also have an impact on the mortality of cancer patients [7,8,9]. Consequently, financial toxicity can directly impact health outcomes, as recognised by Carrera et al. [10].

Based on the framework by Witte et al. [4] (see Figure 1), subjective financial distress consists of three domains with six subcategories:(a)Material conditions: impact of individual financial spending and financial resources used (e.g., more expenses than expected, use of saving to cover spending)(b)Psychosocial response: individual perception and psychological consequences of financial burden (e.g., concerns or worry about financial situation)(c)Coping behaviour: personal approach to cope with financial burden (e.g., using financial assistance or reducing leisure activities)

Until today, most studies on financial toxicity were conducted in the US, so that the majority of validated instruments refer to the US setting. Overall, the prevalence of financial toxicity among cancer patients in the US ranges from 39 to 64% [11], whereas a recent review by Longo et al. [12] discovered a prevalence of 7 to 39% in publicly funded healthcare systems. To some extent, the difference relates to the design of the healthcare system in the US without fully implemented universal healthcare coverage and a less exhaustive social security system, which could foster financial toxicity. Besides systemic factors, an exhaustive literature review by Smith et al. [11] from the US setting identified several socioeconomic, demographic and clinical variables as the main risk factors of objective and subjective financial distress leading to financial toxicity. Additionally, other reviews without limitations of country settings identified sociodemographic and especially employment-related risk factors [13,14], but they were driven by included studies from the US.

As acknowledged by several authors [4,11,12,15], the transferability of risk factors to other high-income countries with different healthcare systems seems questionable because different aspects might come into play. Due to global trends of prolonged survival and improved diagnostics, an increase in the number of younger cancer patients and thus, patients at risk of financial toxicity, is further expected. It adds additional strain to the system besides already occurring developments of rising costs for innovative anticancer drugs. Therefore, an individual analysis of risk factors for countries with universal healthcare coverage is needed, as the determination of them contributes to a comprehensive understanding about the occurrence, degree and extent of financial toxicity. It could benefit the development of effective interventions by identifying patients at a greater risk of experiencing financial toxicity. Moreover, as the currently used instruments to measure financial toxicity are mainly designed for a US-context, finding domains of risk factors that are important in other contexts could inform the evolvement of precise measurements in countries with universal healthcare coverage.

Objective financial burden in publicly funded healthcare systems in terms of (in-)direct costs have largely been quantified, depending on the cancer entity and degree of healthcare coverage [12,16,17,18]. For instance, Longo et al. [12] identified out-of-pocket (OOP) costs and income losses as main contributors to financial toxicity based on the authors understanding of financial toxicity as a consequence of objective financial burden. A recent review of qualitative studies in publicly funded healthcare systems by Fitch et al. [15] supports sociodemographic variables, especially pre-diagnosis financial circumstances, OOP burden and individual ability to apply coping strategies as predictors for financial toxicity accounting for both objective and subjective financial distress. Both studies aim to analyse the structure of financial toxicity rather than evaluate underlying risk factors. Moreover, most reviews were biased by the understanding of financial toxicity as a consequence of objective financial burden, so that only an inconclusive picture of subjective financial distress in universal healthcare settings exists.

Therefore, this review aims to answer the underlying research question of which contributing risk factors for experiencing subjective financial distress due to cancer occur in universal healthcare coverage settings of high-income countries. Besides identifying relevant risk factors, the aim is to determine the time-dependent occurrence of the risk factors for subjective financial distress before, during and after the course of disease to contribute to a better understanding of the prevalence and structure of risk factors. The review will inform the evolving research of appropriate and valid instruments to measure financial toxicity as well as effective responses to tackle the issue in third-party payer systems.

## 2. Methods

### 2.1. Information Sources, Search Strategy and Study Selection

A systematic literature search was performed in the databases of PubMed, PsycINFO and CINAHL until December 2020. The review was conducted following the Preferred Reporting Items for Systematic Reviews and Meta-Analyses (PRISMA) statement [19]. The search strategy was collaboratively developed combining synonyms and related terms of cancer and financial difficulties based on adjusted taxonomies of financial toxicity [4] (for a detailed search strategy see Appendix A). Reference lists of identified systematic reviews and included studies were manually screened for any other eligible studies.

Title and abstract screening as well as full-text screening were performed independently and simultaneously by two authors (SP and BS) based on pre-defined criteria. All dissents were resolved by consensus.

### 2.2. Eligibility Criteria

Articles on cancer patients’ risk factors for subjective financial distress were included, in which risk factors were directly linked to the patients’ perception of their financial situation. The population of interest was adults who were at least 18 years old at diagnosis. No restrictions were placed on cancer entity, stage of disease, employment status, timing of financial distress assessment or research method used (quantitative/qualitative studies). In order to increase comparability of results at the healthcare system level, studies had to be conducted in high-income countries, as classified by the World Bank [20], and with universal healthcare coverage, per definition by the WHO [21].

Study settings of clinical trials evaluating specific therapies were excluded in order to assess the current status quo of financial distress in routine care. To identify individual risk factors of patients experiencing financial distress, studies from the perspective of caregivers, family members or systemic stakeholders were excluded. Studies solely assessing OOP costs, productivity losses or return to work situations without linkage to subjective financial distress were also excluded. Publications were restricted to the English or German language.

### 2.3. Quality Appraisal

In order to determine the validity and quality of eligible studies, a risk of bias assessment was performed by two reviewers (S.P., B.S.) using two different checklists to account for the included qualitative and quantitative studies. For the latter, the NIH Quality Assessment Tool for Observational Cohort and Cross-Sectional Studies with 14 items [22] was applied, as it accounts for the internal validity of the evaluated exposures. It comprises a subjective quality rating (good, fair or poor) of the individual papers, which was quantified based on the numbers of positive assessed quality criteria (good: met all applicable categories; fair: most categories and low: minority of categories). Risk of bias for qualitative studies was assessed using the Critical Appraisal Skills Programme (CASP) Checklist for qualitative studies [23], which focusses on the validity of collected results.

### 2.4. Data Extraction and Qualitative Synthesis

Data extraction and synthesis was split between two authors (S.P. and B.S.). Disagreements and indecisions were resolved by group discussions and consensus throughout a multidisciplinary research team (oncology/medical ethics: K.M., A.Z., E.C.W., social sciences: L.R., N.M. and health economics: S.P., B.S., W.G.). To account for possible heterogeneity in measurement of financial toxicity across studies, the underlying definitions of subjective financial distress were categorised into the dimensions (material, psychosocial and behavioural) of the framework of financial toxicity by Witte et al. (2019) [4]. To account for country-specific healthcare systems (public, private or societal financing) and related individual copayments (voluntary insurances and out-of-pocket spending), countries were characterised by their health expenditure and type of financing based on the most recent OECD data [24].

To account for heterogeneity in studies (e.g., healthcare systems, measures of financial distress, statistical methods) a qualitative approach to data synthesis was chosen. Applying an inductive approach of a qualitative content analysis following Mayring [25], categories and subcategories were derived from the risk factors identified in the studies. Each (sub)category was evaluated with regard to its time-dependent occurrence in the course of the disease, and a cause–effect model of risk factors for subjective financial distress was evolved. Time-dependency was categorised into “before cancer diagnosis”, “during cancer treatment” and “after cancer”, the latter being defined as cancer survivors as per the definition in the respective study or more than five years after treatment, assuming that a patient is counted as cured, parallel to clinical practice [26]. For the identified categories and their risk factors, it was assessed whether a possible effect on financial distress was tested for in a quantitative study and, if so, whether they reached statistical significance. The effects and their statistical significance were extracted from the analyses. All reported risk factors were classified based on the reported direction of effects as either “promotive”, meaning to contribute to subjective financial distress, or “preventive” indicating to prevent the patient against the distress.

## 3. Results

A total of 4321 records were identified until December 2020, of which 284 full texts were retrieved after duplicates were removed and titles/abstracts were screened. 46 papers matched the pre-defined inclusion criteria and were included (see Figure 2). Most full texts were excluded during the screening process due to their country setting (mainly US, *n* = 141), differing study objective (*n* = 43) or publication type such as editorial or review (*n* = 42).

Included studies are 30 quantitative studies based on surveys and 16 qualitative studies using either individual/group interviews (*n* = 13) or answers to open-ended questions in surveys (*n* = 3) (see Table 1. For a comprehensive overview of study characteristics see Appendix A). The studies were mainly conducted in Europe (*n* = 27) with most quantitative studies from Germany (*n* = 8) [7,16,27,28,29,30,31,32] and most qualitative studies from the United Kingdom (*n* = 5) [33,34,35,36,37]. Across both study types most studies were performed in Australia (*n* = 11). Included healthcare systems are predominantly based on either general tax revenues (Norway, United Kingdom, Italy, Ireland, New Zealand, Australia, Canada and Finland) or compulsory health insurance schemes (Lithuania, Germany, Japan, Netherlands, France) with different degrees of personal contribution (voluntary health insurance and out-of-pocket costs) ranging from 33% in Lithuania to 14% in Germany and Norway [24] (for a comprehensive overview see Appendix A).

The studies consider a variety of participant characteristics, with most study samples nearly equally distributed between men and women, although one study only considers men [38] and six studies only women [30,31,32,39,40,41]. Different cancer entities are distributed across studies with primarily gynaecological (*n* = 26), gastrointestinal (*n* = 21) and urogenital (*n* = 18) cancers. Most studies recruited patients with pre-defined cancer entities while some studies did not restrict the recruiting for any cancer entity [16,42,43,44,45,46,47] nor did they report the entities of the sample [33,34,35,36,37,48]. A variety of different time points during and after the course of disease are scrutinised in the included studies. Most studies interviewed the included patients shortly after their diagnosis <2 years (*n* = 21), while a total of 21 studies do not report sample characteristics of or restrictions made for time since diagnosis, especially within qualitative studies (*n* = 13).

### 3.1. Measuring Subjective Financial Distress

Subjective financial distress was consistently recognised as a multidimensional construct. The operationalisation of subjective financial distress, however, differed across studies regarding (1) instruments, (2) domains and (3) statistical approaches. In most studies (*n* = 28), financial distress was assessed by a subscale of a patient-reported outcome measure with a different focus (often HRQoL), whereas six studies [7,43,49,50,51,52] used specific instruments for financial distress. The majority of instruments assess financial distress through questions regarding material conditions, especially when applying validated instruments (*n* = 21). Studies using validated subscales usually used the EORTC-QLQ30 questionnaire (*n* = 14), in which financial difficulties are measured by a single item (“During the past week: Has your physical condition or medical treatment caused you financial difficulties?”). Studies with full instruments specially developed to measure financial distress mainly referred to the 11-item COST measure (*n* = 4), which was originally validated in US patients [53]. Table 2 provides an overview of the types of instruments that were applied and the domains of financial distress they address.

Eleven studies applied non-validated subscales, often incorporated within self-designed instruments and mainly measuring objective financial burden (e.g., OOP costs). Psychosocial responses were relatively more integrated in non-validated subscales, either complementary [38,50] or substitutive [46,54,55,56,57,58] to material components. Behavioural aspects were less often considered, except for [7,38,50,59]. 

While material financial burden dominates the notion of financial distress within qualitative interviews (*n* = 10), qualitative studies often acknowledge a broader understanding of subjective financial distress, addressing several domains depending on the underlying research question (e.g., financial concerns of rural breast cancer patients [40] illustrating the psychosocial domain).

In order to assess the impact of potential risk factors on financial distress, different approaches are pursued in quantitative studies. Some studies report the measured frequencies of identified prevalent risk factors [16,32,38,50], while others tested for differences between groups or time points using null-hypothesis significance testing [27,28,31,54,60,61,62]. Most studies (*n* = 20) controlled for potential confounders and evaluated predictors for financial distress through regression models or analyses of variance. The included studies assessed the perceived financial distress within a sample of cancer patients, while three studies [29,30,31] also determined the extent of financial distress in comparison to the general public.

### 3.2. Quality Appraisal

The included quantitative and qualitative studies are mainly rated fair on overall quality (for a more detailed quality appraisal, see Appendix A). Across studies, the main risk of bias appears to be a possible selection bias of included participants, as most studies used a non-probabilistic sampling technique, except for [39,42]. Retrospective and cross-sectional study designs (*n* = 38) can also imply recall bias due to a patient’s individual assessment of subjective financial distress in the past. Some papers refer country-specifically to the same data, namely, two qualitative papers from Ireland [63,64], two quantitative papers from France [54,55] and five from Germany [27,28,29,30,31], which might lead to an overestimation of risk factors.

Most quantitative studies were assessed as having fair quality, although one study was considered to be of low quality [54] and one of good quality [39]. Most studies defined the population of interest well, even though some studies were limited by imprecise population samples due to inconsistently applied eligibility criteria [32,42] or different time periods considered [27,28,30]. Except for two studies [38,39], sample size justifications were lacking. Exposures were clearly defined and measured in all studies, but the outcome measure was limited by ten studies [7,38,45,46,54,55,56,58,62] using non-validated instruments for evaluating financial distress. Six studies [16,38,50,54,60,62] did not statistically control for confounding variables regarding the outcome of financial distress.

The qualitative studies demonstrated fair quality, with seven studies only missing one criterion [33,34,63,64,65,66,67]. All studies show sufficient quality in terms of underlying research method and data collection. Limited quality occurred regarding analysis of potential bias, mainly in terms of relationship between researcher and participants [33,34,35,36,37,40,41,48,63,64,65,66,67,68] and sufficient data analysis [35,36,37,41,48,68].

### 3.3. Risk Factors for Experiencing Financial Toxicity

All studies that compared cancer patients to the general public indicate a significantly higher financial burden for individuals with cancer [29,30,31]. Despite different underlying definitions and measurement methods, the results are predominantly consistent across studies with regard to identified risk groups for subjective financial distress.

#### 3.3.1. Categories of Risk Factors and Their Prevalence

A total of 585 risk factors were extracted from the studies, although some risk factors were overlapping and similarly reported in several studies. Based on the qualitative content analysis, these risk factors could be classified into eight categories with a total of 34 subcategories, each category ranging between two and seven subcategories (see Figure 3). The most comprehensive category is sociodemographic factors, such as age, education, familial status, gender and socioeconomic status, which were studied by 25 out of 46 included studies especially in terms of age (*n* = 18) and familial status (*n* = 15). Within this category, country relates to the specific country of living while geography refers to whether the patient lives in an urban/rural region and/or has to travel a lot to receive treatment. Categories of employment (*n* = 25), financial resources (*n* = 23) and social circumstances (*n* = 14), which could generally be assigned to sociodemographic factors, were given their own category in order to account for their special significance within the financial distress framework. This was also justified by their prevalence within the studies. The categories of employment and financial resources were driven by factors occurring before the diagnosis either in terms of the work status (*n* = 20) or the financial status (*n* = 14) Financial role within social circumstances refers to the extent of individual contribution to the financial situation within the family (e.g., being the main bread winner). Directly medical and treatment-related factors of the cancer disease were also recognised in 25 studies, either in terms of the time since onset of disease (*n* = 10), indicating time between diagnosis or treatment of cancer and the time point of measurement, or regarding the treatment (*n* = 12) chosen. The category of insurance comprises country-specific systemic factors about utilisation of health insurance and social security systems, while knowledge implies personal knowledge about access to care and benefits as well as their past experiences with these systems. Especially the subcategory of health insurance was mainly studied (*n* = 16) across the papers. A comprehensive overview of all financial distress related risk factors and their assigned categories can be found in Appendix A. 

#### 3.3.2. Incidence of Risk Factors during the Course of the Disease

The identified (sub)categories of risk factors (see Figure 3) occur at different stages during the course of the disease and contribute to subjective financial distress. Based on the time points measured and questions asked within the included studies, time-dependent occurrence of the risk factors could be derived. The time points were transferred on an aggregated level of their related (sub)categories and set in relation to one another, as illustrated in Figure 4. Time as a risk factor, as the time between the occurrence of cancer and the measurement of financial distress, occurs during the entire course of the disease, even beyond survivorship [16,28,30,31,38,43,47,54,60,61].

The individual situation before the disease, often in terms of sociodemographic characteristics [7,29,31,34,40,41,42,43,44,45,46,47,49,51,52,54,55,56,57,58,60,61,62,64,69,70], shapes the degree of risk factors and the outcome of financial distress during and after cancer. Pre-diagnosis financial status [36,43,44,49,51,52,54,57,64,65,68] and work status [7,33,35,36,39,40,41,43,44,45,47,58,64,65,68,69,71] influence changes in income [36,52] and ability to work [65,67] during the disease, while the latter also lasts into the degree of return to work after disease [27,32,47,63]. Besides sociodemographic factors, past experiences [33,36,37,71] and knowledge [34,36,37,71] about systematic factors and cancer itself [33,36] also matter during illness.

Overall, risk factors before and during disease are shaped by systemic factors, such as the design of health insurance [7,33,34,36,40,43,44,49,52,55,58,64,65,67,68,71] or social security system [34,36,48,64,65,66,67]. Medical factors, such as cancer entity, cancer stage [38,43,44,45,47,54,55] or chosen treatment [7,39,44,45,49,54,55,56,57,60,64,67,68] determine the severity of illness and have an influence on the degree of financial distress. This affects associated side effects [37,40,47,48] and experienced well-being [39,46,47,54,55] during the entire course of the disease. The disease and treatment cause direct (non-)medical costs not covered by health insurance [49,58,69]. They lead to coping behaviour in which patients make life changes to adapt to the situation, such as cutting down leisure activity [36] or using savings [35,38,56,62,65,67,71]. Besides personal coping mechanisms, patients also refer to institutional support (e.g. social workers) [34,36,37,48,65,66,67,71] depending on previous individual knowledge and beliefs about assistance [33,36,37,40,71]. Family support is sought continuously during the course of the disease beginning with psychological support [34,35,52,56,67,71] at diagnosis or borrowing money [35,36,64,65], if financial burden exists. 

Fewer individual risk factors occur after the disease, so that the risk of subjective financial distress is built on the success of treatment to survive the disease. If survived, the degree of ability to return to work [27,32,47,63] and possible adjustments [27] made as a consequence of disability through cancer disease influence late financial distress. 

#### 3.3.3. Quantified Influence of Risk Factors

Out of 325 risk factors identified in quantitative studies, 140 showed significance within their study. In general, significant risk factors were often related to (sub)categories of (sociodemographic) factors before the disease or to clinical and work-related factors, whereas psychosocial or systemic categories (e.g., social circumstances, insurance, knowledge) were never or only very rarely significant or not measured in the included quantitative studies. Table 3 gives an overview of the subcategories that have a significant impact on financial distress.

Most risk factors were consistent across studies in terms of direction of effect, whereas some, such as age and gender, showed inconsistent results, which may be attributed to differences in study populations. For age, it appears that an age below 65 years [31,45,49] promotes financial distress, while increasing age [43,47,51,54,69] and age above 64 years [42,57] prevents it. For gender, one study identified men [47,58] and two studies women [7,54] as being at greater risk for financial distress.

Being alone (single, divorced or separated) [46,54,55] and having dependants [56,58] contribute to financial distress, while living with the family [45,69] or being married [47,69] in general counteract it. Pre-diagnosis financial status in terms of lower income [54,58,62] and pre-diagnosis financial stress [57,58] could promote whereas having savings and private health insurance [7,44,49,58] could prevent financial distress.

Clinical factors are recognised as another driver of financial distress. Koch et al. [31] demonstrate an increase in financial distress shortly after diagnosis (<1 year) and longer time since diagnosis (>5 years), while having decreased in between. Generally, (very) long-term survivors (>5 years) [28,30,31] had greater financial distress than their comparators. An in-depth comparison by Doege et al. [30] further indicates more financial distress within very long-term survivors vs. long-term survivors, but the result is not significant. Besides cancer status and treatment, impaired well-being in terms of increasing scores of depression and anxiety [46,55] through HADS and ESAS as well as lower functioning levels [46] of FACT-G (all validated instruments) lead to greater financial distress [47,55].

Total (OOP) costs [49,69], high medical costs [58] and increased household bills [58] are quantified as promotive risk factors while lower direct medical costs [58] are preventive of financial distress. Coping behaviour as either a direct consequence of OOP burden [43] or through unspecified usage of savings [56] also indicates a promotive contribution.

Besides monetary implications, impacts of disease and treatment on employment also predict financial distress depending on the work status before diagnosis. Changes in work status because of cancer (unemployment, retirement, disability) [39,43] during the course of the disease do promote subjective financial distress, while the previous work status, regardless of the type (employment, unemployment, retirement) [47,58], prevents against it. The mode of return to work as reducing working hours or the uptake of a new job, also has a promotive effect [27,32].

A sensitivity analysis, in which all studies with a sample size smaller than 100 [44,51,60,61] were excluded, revealed that all significant effects found in these studies were also confirmed through studies with larger sample sizes. Therefore, excluding these studies did not alter the above results.

While several risk factors were included in quantitative articles, some risk factors revealed from qualitative studies have not yet been measured and tested in quantitative studies. Most unquantified risk factors focus on personal situation beyond sociodemographic factors, such as financial commitments before diagnosis [41,52,64,68], social circumstances of the patient [35,36,41,52,64,65,68,71] and direct implications of employment [33,35,41,48,52,63,65,71] or income losses [35,36,37,48,52,63,64,65,66] during the course of the disease. Several behavioural aspects of coping with the situation, such as economising on household’s expenditure [35,36,67], were mentioned. Furthermore, systemic factors of patient’s country of living were identified across qualitative studies in terms of receiving institutional support (healthcare and social security) [36,48,64,67] and work design [35,36,52,65,71]. Personal knowledge and attitudes about the country’s system [34,36,37,71] have been extensively described as risk factors without being quantitatively evaluated yet. A comprehensive overview of risk factors that are not evaluated for their applicability in a pre-defined target population is given in Table 4. 

## 4. Discussion

This literature review provides a comprehensive overview of the landscape of risk factors for experiencing subjective financial distress in high-income countries with universal healthcare coverage. The results of this review indicate that the experience of subjective financial distress strongly depends on individual sociodemographic factors (e.g., family status, education) and especially on financial status before the disease influencing the degree of financial resources used (e.g., savings) to mitigate the financial burden during the disease. Work-related factors such as work status before the disease and returning to work after the disease also have a strong impact on subjective financial distress [5]. The distress is also determined by the time since onset of the disease and the design of healthcare as well as the social benefit system. Qualitative papers also indicate that limited knowledge of patients regarding access to care, cancer illness and potential financial toxicity due to cancer are influential, but these factors need to be quantified in further studies. 

The different approaches to the measurement of subjective financial distress observed in the reviewed studies indicate that a generally accepted definition of subjective financial distress is still lacking and that knowledge concerning underlying risk factors is limited. The various (methodological) approaches, as well as different focuses in assessing risk factors, rendered the comparison of the included studies difficult and gave the results of this review an exploratory character. The range of identified risk factors suggests that these risk factors are a multidimensional construct. To better understand this multidimensionality, further research should focus on the entanglement of the individual risk factors and how they depend on each other.

As many of the identified risk factors are directly associated with the diagnosis of cancer and/or its treatment, cancer itself elevates the risk of financial distress. Subjective financial distress is not only triggered by direct or indirect costs for cancer care, but also by a number of additional individual risk factors that manifest themselves, for example, in the form of personal pre-conditions, behavioural patterns and coping strategies. Due to these components and the requirement of an individual assessment regarding the financial distress, subjective financial distress needs to be handled as a multidimensional patient-reported outcome (PRO) in order to address and alleviate financial distress. Such PROs will further help to assess which factors are relevant in specific patient groups. The results of this review could provide the basis for the development of an instrument for measuring subjective financial distress, as it systematically maps the landscape of existing risk factors that could be operationalised as surrogates to measure the degree of subjective financial distress. A standardised PROM is also necessary to determine the relevance of risk factors and the frequencies with which they occur. It can be used to inform treatment choices based on financial considerations or to profile patients and to detect patients at risk for financial distress and with a special need for financial support and for counselling in this regard. The development of financial support programmes and informing decision-making on policy and regulatory changes could be further potential applications for such an instrument.

Although sociodemographic factors and related pre-diagnosis financial status are important determinants of subjective financial distress, these risk factors, however, are mostly static and contain limited potential for change in the short term, as they are built over years. Since access to healthcare and social benefits also varies significantly between sociodemographic groups and as limited knowledge of how to access to care and benefits promotes subjective financial distress [34,36,37,71], the navigation throughout the system as well as the underlying bureaucratic obstacles should be simplified. Other risk factors related to return to work should also be addressed by policy making, but this requires rethinking on several levels. To mitigate the risk of this factor, the exchanges between third-party payers, employers and patient representatives should be strengthened, focusing on vocational reintegration. To tackle the reasons for a hindered or a successful return to work, further research, such as the study by Janßen et al., 2001 [72], is needed. Such long-term effects illustrate the transition of cancer into a chronic disease and provide indications of where adjustments can be made in healthcare policy making to reduce the financial distress in cancer patients.

Concerning the factor time since disease onset, the risk for financial distress is mainly occurring at a shorter (<1 year) [31] and longer time (>5 years) [27,30,31]. Thus, there is a need for a consulting and support service at an early stage of the disease, e.g., by establishing early screening tools in practice. Moreover, a forward-looking consultation is indispensable. Raising awareness regarding potential financial toxicity should be prioritised in an early stage of the disease, too, as limited knowledge might promote subjective financial distress [33]. As financial toxicity could affect health outcomes, the communication of financial toxicity as a common side effect should be more encouraged in clinical consultations. The risk of experiencing subjective financial distress as a cancer survivor requires a healthcare and social benefit system that is supportive not only in acute but especially in chronic diseases. 

Other exogenous factors concerning the design of healthcare and social security systems might also shape the extent of not only objective financial burden but also protection of vulnerable cancer patients at risk of financial distress. Several studies displayed the preventive effect of private health insurance [7,44,49,58] and some studies evaluated country-specific healthcare assistance (e.g., receiving a medical card in Ireland) [58]. Qualitative studies went beyond that and identified social welfare provisions [34,36,48,64,65,66,67] as well as the regulation of employment (e.g., sick leave, employer benefit schemes) [34,35,51,64,70] as important stressors, again illustrating the importance of work-related risk factors and indicating further parameters that can be addressed by policy makers to alleviate the financial burden of cancer patients. Furthermore, while exogenous risk factors such as design of health insurance or employer benefit schemes are very country-specific, we recognised that the aggregated (sub)categories occur equally across countries. The risk factors affecting the individuals themselves, however, do not show any meaningful differences across jurisdictions, even though the personal contribution to healthcare ranged from 33% in Lithuania to 14% in Germany and Norway [24]. It indicates that the main (sub)categories for experiencing subjective financial distress are equally applicable to universal healthcare settings, but the degree of subjective financial distress is shaped by country-specific provisions. Thus, besides country-specific research and interventions needed to tackle the rising side effect of financial toxicity, it should be discussed whether the development of country-specific instruments might be beneficial in order to account for systematic factors. Progress in the development of appropriate country-specific instruments for measuring financial toxicity can already be observed in research, e.g., in Italy [73] or Canada [74].

Previous reviews on financial toxicity considering third-party payer systems also identified sociodemographic characteristics and time since onset of the disease as the main driver of financial toxicity [12,13,14,15]. They, however, identified being younger than retirement age [12,14] and being female [14] as risk factors, whereas our review does not provide consistent results on this. The inconsistency across the reviews might result from a different scope of reviews, as Longo et al. [12] consider cancer patients with less than five years since diagnosis while Gordon et al. [14] analyse cancer survivors including studies up to January 2013, whereas our review included studies of cancer patients without restrictions on the disease stage. For age, several studies indicate a preventive effect of increasing age [43,47,51,54,69] and being above retirement age (>65 years) [42,57], whereas two studies identified older patients at risk [31,46]. Since the studies showing a preventive effect of increasing age applied justified multivariate regression models, the value can be rated higher than the opposite results of the other two studies, where either no regression model was applied [31] or the regression model itself provided inconsistent results [46]. A preventive effect of increasing age and above retirement age seems also plausible, as, e.g., their ability to save money over time increases and those retired could refer to a continued pension while often being exempt from copayments. The effect of gender was not significant in four studies [44,45,51,60] and inconsistent within three studies that found significant effects [7,47,54,58] but mainly trending towards females being at greater risk.

Comparing the results of this study with equivalent reviews from the US shows that most risk factors overlap in countries with universal healthcare and the US. However, the extent to which the risk factors influence subjective financial distress is different between these countries. In universal healthcare, work-related risk factors are more important than OOP costs, so risk mitigation strategies may differ from those in the US. Smith et al. found that in the US, lower adherence to treatment to save medical costs is a consequence of financial burden [11]. This was not observed in our study. Nevertheless, it is important to note that treatment-related avoidance strategies [38,50,67] and behavioural aspects [32,33,34,35,36,41,50,54,65,69] have generally been studied only to some extent and mainly in qualitative studies, so this study could not fully capture the impact of such coping behaviour as a risk factor. Hence, the analysis of mitigation strategies and their impact on financial distress should be elaborated and incorporated in suitable measures.

This review is subject to various limitations. To mitigate the limitation of comparability and generalisability due to different instruments used for measurement, we only included studies from similar settings in high-income countries with universal healthcare coverage. Still, the various approaches to the measurement of financial distress in the individual studies (especially through measurement with differing instruments) as well as the use of different statistical methods yielded in very heterogeneous results that are difficult to compare with each other. Thus, direct comparisons and/or pooling of effect sizes had to be refrained from, and only the directions of the effects found could be compared. In some studies, it was not reported how the tested risk factors were selected and whether non-significant risk factors were left out of the quantitative analyses. This information would have been helpful for this review to assess whether these risk factors have no influence on financial distress.

Due to a lack of a consistent definition of financial distress, the search strategy used may not have been sufficient to identify all relevant studies on this topic. To counteract this, the bibliographies of the included studies were searched for further relevant articles. Data extraction and synthesis was always performed by at least two authors. Nevertheless, subjectivity in the creation of categories and the time-dependent occurrence model cannot entirely be ruled out. Exclusion of studies that solely assessed the objective burden of OOP costs may have resulted in an underestimation of them as a risk factor, even though the included studies that did consider OOP costs suggested that they did not have a major impact on financial burden.

## 5. Conclusions

In summary, this review filled the gap of an exhaustive determination of associated risk factors for subjective financial distress in cancer patients in high-income countries with universal healthcare coverage. Overall, subjective financial distress is primarily determined by pre-diagnosis sociodemographic, financial and work factors changing during the course of the disease, so that previously sociodemographic disadvantaged and working-aged persons are at a greater risk of experiencing financial toxicity. The design of healthcare and social security systems shape the country-specific degree of subjective financial distress. This review identified further revealed risk factors across qualitative studies, such as mitigation strategies to counteract financial burden and behavioural aspects. These risk factors should be quantified in upcoming studies to determine the degree of influence for financial distress. 

To account for systemic differences, a multidimensional instrument measuring subjective financial distress in universal healthcare settings should be developed by incorporating the identified categories of risk factors as subdomains. This will support an exhaustive determination of the degree and composition of financial distress in countries with universal healthcare coverage which should be studied more in-depth. It is also apparent that financial distress can occur even in countries with universal healthcare, and as the prevalence of cancer increases, financial distress can also expected to increase. The development and implementation of multidisciplinary intervention schemes should be prioritised by national healthcare decision makers. The development of effective schemes should consider the identified risk factors in order to address patients at higher risk of experiencing financial distress more appropriately. 

## Figures and Tables

**Figure 1 cancers-13-05015-f001:**
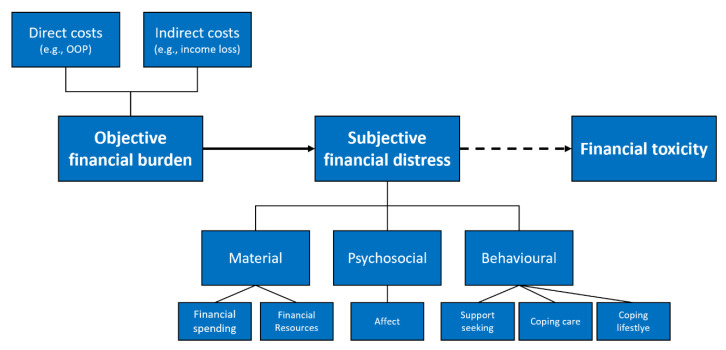
Framework of financial toxicity by Witte et al. [4].

**Figure 2 cancers-13-05015-f002:**
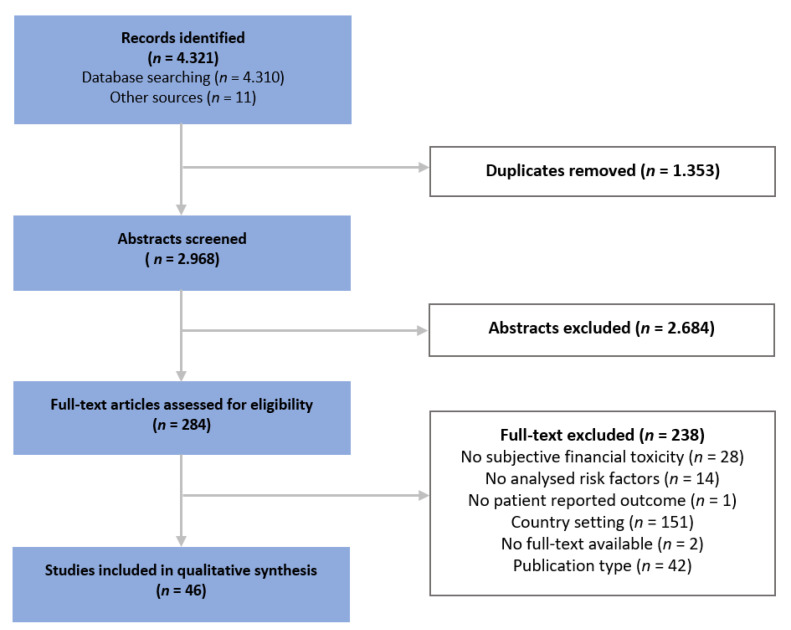
PRISMA flow chart of study selection.

**Figure 3 cancers-13-05015-f003:**
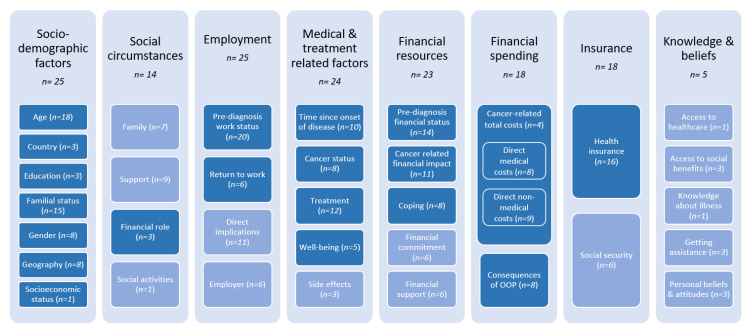
Identified categories of risk factors for financial distress. Categories which showed significant results in quantitative studies are coloured in darker blue. Prevalence of (sub)categories were counted on study level.

**Figure 4 cancers-13-05015-f004:**
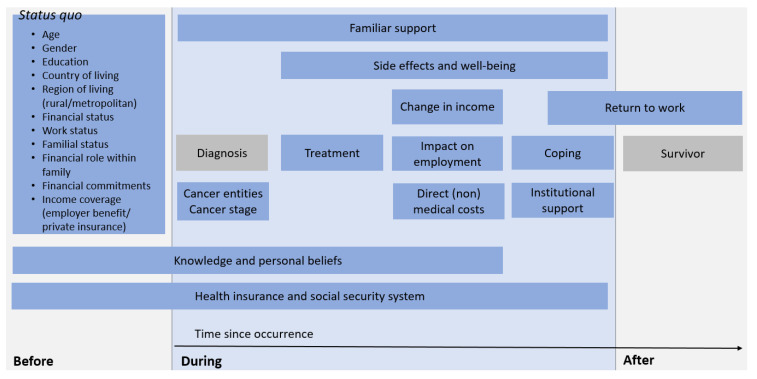
Time-dependent occurrence of identified risk factor (sub)categories for financial distress. Grey coloured risk factors refer to the main stages of the disease pathway (diagnosis and survivorship) accounting for the different origin of the study population experiencing financial toxicity in the included papers.

**Table 1 cancers-13-05015-t001:** Study Characteristics.

Characteristics	Quantitative Studies (*n* = 30)	Qualitative Studies (*n* = 16)
Study design Cross-sectional study Cohort study	23 7	15 1
Continent setting Asia Australia/New Zealand Europe North America	3 8 19 1	0 4 8 4
Cancer entities Gastrointestional Gynaecological Head and neck Haematological malignancies Skin Thorac Urogenital Other Unspecified	17 17 8 5 3 8 12 7 0	4 9 0 2 0 6 6 0 6
Time since diagnosis 0–2 years after diagnosis 2–5 years after diagnosis >5 years after diagnosis Unspecified	21 4 5 8	2 1 0 13
Sample size Minimum Median Arithmetic mean Maximum	43 278.5 915.3 8830	8 22.5 80.75 378

Note: Sample size calculation based on included cancer patients at baseline. For country setting, cancer entities and time since diagnosis, frequencies refer to total number of reported countries/entities/time points within the studies and do not sum up to total number of included studies. Unspecified cancer entities/time since diagnoses were considered, when studies do not report these characteristics (methods/result section).

**Table 2 cancers-13-05015-t002:** Definition of and instruments for financial toxicity.

Definition Instrument	Material	Psychosocial	Behavioural	Not Reported
Full questionnaire (*n* = 6) Validated Non-validated	5 1	5 1	0 1	0 0
Subscale (*n* = 28) Validated Non-validated	15 6	0 9	0 3	0 0
Interviews (*n* = 13) Individual interviews Focus groups	10 0	5 1	5 0	2 0

**Table 3 cancers-13-05015-t003:** Effects of quantified risk factors.

Categories	Subcategories	Promotive	Preventive
Sociodemographic factors	Age	Age < 65 [31,45,49] Age > 65 [31] Increasing age (until 60 years) [47] Increasing age [46,47]	Greater Age [55] Age > 64 [42,57] Increasing Age [43,47,51,54,69]
	Country	Depending on country of living [61]	
	Education		High Education [69]
	Familial status	Being single [54,55] Being divorced or separated [46] Having one or more dependants [56,58]	Living in a family home [45] Being married, living with partner [47,69]
	Gender	Male [47] Female [7,54]	
	Geography	Living in a major city [62] Living away from treatment center [70]	
	Socioeconomic status		Medium or high socioeconomic status [47]
Social circumstances	Support	Support by family and friends [56]	
Medical and treatment related factors	Time since occurrence	Short time since diagnosis (<1 year) [31] Longer time since diagnosis (>5 years) [31] Long-term survivor (5–9 years) [30,31] Very long-term survivor (>10 years) [28,30,31]	Medium time since diagnosis (3 years since diagnosis) [31]
	Cancer Status	Metastatic cancer [54,55] Cancer entity [47]	
	Treatment	Combination therapy (surgery + X) [39,57], Surgery [55] Stoma [56]	Undergoing psychological counselling [39]
	Well-being	Comorbidities > 2 [39] Higher scores of HADS or ESAS [46,55] Poor social/family well-being [46]	Low HADS-D score [55] High emotional, cognitive, social functioning [47,55] Better functional and social well-being [55]
Financial resources	Pre-diagnosisfinancial status	Lower income [54,58,62] Less income in metropolitan region [62] Pre-diagnosis financial stress [57,58]	Having savings [43]
	Financial impact	Objective financial burden [7,54] Subjective financial difficulties [54,59]	
	Coping	Using up savings [56] Having no savings to use up [56]	
Financial spending	Cancer-related total costs	Total costs [69] OOP costs [49,69]	
	Direct medical costs	High direct medical costs [58]	Low direct medical costs [58]
	Direct non-medical costs	Increased household bills [58]	
	Consequences of OOP	Coping strategies used to cover expenses [43]	
	Financial commitment	Having a mortgage and/or personal loan(s) [58]	
Employment	Return to work	Change in: - Working hours [27] - Leaving former employment [27,32]	
	Pre-diagnosis work status	Change in work status because of cancer: - Becoming unemployed [39] - Retiring [39,43] - Disabled [39] Working part-time [43]	Work status before diagnosis: - Having paid employment [47] - Not working [58] - Being retired [58]
Insurance	Health insurance	Obtaining healthcare assistance after diagnosis [58]	Private health insurance [7,44,49,58]

**Table 4 cancers-13-05015-t004:** Not Evaluated Risk Factors in Quantitative Studies.

Categories	Subcategories	Risk Factors
Social circumstances	Family	Distress through children at home or in education [36,65] Illness of further family members [52] Work adjustments of carers [35] Social isolation [64]
	Financial role	Cancer patient as previous main earner of family [36,41,68]
	Social activities	Reduction in leisure activities [36]
	Support	Financial support by family members [35,36] Community and NGOs’ support [68,71]
Financial resources	Pre-diagnosis financial status	Income of partner [36]
	Financial commitment	Home owner/mortgage [41,52,64,68] Repayments of credits [68] Expenses for dependant children [33]
	Financial support	Contributions through social security system [64,68] Receiving sick pay [33,64] Coverage of private insurance [36,68,71]
Financial spending	Consequences of OOP costs	Economising on household expenditure [35,36] - Expenses for daily living [33,65,67] - Transportation [67] - Major purchases [67] - Medical remedies [67] - Holidays [67] - Leisure acitivities [67]
Employment	Return to work	Return to work during disease [65,67]
	Direct implications	Changes in employment during disease [33,35,41,48,52,63,65,71] Loss/reduction in income [35,36,37,48,52,63,65,66] Changes in partner’s employment [63,65,71]
	Employer	Employer benefit scheme [35,36,52,65,71] Supportive employer [48] Working in public sector [36]
Insurance	Health insurance	Increased insurance premiums [33] Time between travel expenses and reimbursement [52]
	Social security system	Disability coverage [34,65,66] Institutional support [36,48,64,67] Ineligible for social welfare benefits [64]
Knowledge	Access to healthcare	Treatment provider [71] Expenses [71]
	Access to social benefits	Lack of knowledge about available benefits [34,36,37] Dealing with bureaucratic system [34]
	Assistance	Assistance by social worker/welfare rights advisor [34,36] Time between diagnosis and receipt of advice [36,37] Professionals who did not alert patients to benefit entitlements [37]
	Knowledge about illness	Cancer-related financial distress [33] Course of disease [33]
	Personal beliefs and attitudes	Stigma of financial distress [33] Negative attitudes benefit system [36,37] Beliefs about the extent and severity of illness [36]

## Data Availability

Not applicable.

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
