# Peer review of "Patient-Reported Financial Distress in Cancer: A Systematic Review of Risk Factors in Universal Healthcare Systems"

_cancers, 2021, doi:10.3390/cancers13195015_

Round 1
Reviewer 1 Report
The subject of this systematic review is important and timely. The manuscript is very well written, details are given on all the phases of the research and analysis. Figures 3 and 4 are very nice. The Authors acknowledge many of the limitation of their analysis.
I have two major concerns, that represent limitations of the value of the review.
First, variability and uncertainty deriving from the definition of financial distress in the reviewed literature, that is done with different and sometime not validated instruments. This is acknowledged by the Authors and I guess there is no room for improvement. Eventually, in the discussion they might review what instruments are being developed in the countries included in the analysis.
Second, quality of the selected papers. Particularly, I fully agree on inclusion criteria but disagree on excluding papers reporting data from clinical trials (Paragraph 2.2). Clinical trials indeed are the typical setting where prospective patient enrolment is planned and the use of PROs is encouraged and done with some control on the quality. As a consequence, these reports are those that usually have better quality. The explanation given that clinical trials were excluded in order to assess current status quo of financial distress in routine care is not convincing for this reviewer. How many studies were eliminated due to this exclusion criterium?
The consequence of such exclusion is that the quality of the selected papers is mostly fair and affected by a possible selection bias. Such a bias may be even more critical within small retrospective or cross-sectional studies. Sample size is a critical point for quantitative analysis considering that it was justified only in 2/30 studies according to what the Authors report. An exclusion based on some threshold of sample size could have been considered. Actually, the size of the included studies might be very small particularly for the qualitative ones (minimum 8 cases, median 22) but also for the quantitative ones (minimum 43, median 278); how many less than 100 patients?
Some of the above criticisms might benefit of sensitivity analyses (eg. Including data from clinical trials, or excluding data from papers with less than 100 patients…).
The data of the section dedicated to quantified risk factors (para 3.3.3 and table 3) are weak due to quality limitations of the selected evidence papers. In the end, that something is reported as statistically significant in fair quality study probably affected by several selection biases does not mean too much for me. In addition, these data are confused and include statements that are internally inconsistent (as expected due to heterogeneity of the selected papers and acknowledged by the Authors).
Author Response
| Reviewer comments |
Response by authors |
|
The subject of this systematic review is important and timely. The manuscript is very well written, details are given on all the phases of the research and analysis. Figures 3 and 4 are very nice. The Authors acknowledge many of the limitation of their analysis. (R1) |
Thank you so much for your thoughtful and helpful review of our paper. We really appreciated your contribution to our paper. |
|
First, variability and uncertainty deriving from the definition of financial distress in the reviewed literature, that is done with different and sometime not validated instruments. This is acknowledged by the Authors and I guess there is no room for improvement. Eventually, in the discussion they might review what instruments are being developed in the countries included in the analysis. (R1) |
Thank you for sharing your important thought that we fully agree on. We also recognized the limitations through different underlying definitions of financial distress, which we considered within the data extraction process and the analysis later on. As the research on financial toxicity in universal healthcare coverage settings is currently evolving, the existence of appropriated validated instruments for these countries is limited. We hope, that our research could contribute not only to the discussion about the construct definition of financial toxicity, but also to the development process of valid instruments, so that further reviews could deal with more comparable results instead of scattered results as presented. Fortunately, we also recognized ambitions of other research teams across jurisdictions developing national specific instruments, so we added a sentence within the discussion to illustrate the progress in e.g. Italy and Canada, as recommended by the reviewer. |
|
Second, quality of the selected papers. Particularly, I fully agree on inclusion criteria but disagree on excluding papers reporting data from clinical trials (Paragraph 2.2). Clinical trials indeed are the typical setting where prospective patient enrolment is planned and the use of PROs is encouraged and done with some control on the quality. As a consequence, these reports are those that usually have better quality. The explanation given that clinical trials were excluded in order to assess current status quo of financial distress in routine care is not convincing for this reviewer. How many studies were eliminated due to this exclusion criterium? (R1) |
Thank you for questioning our exclusion criteria. We have clarified the description of the exclusion criteria regarding clinical trials to make it clear that we mean clinical trials evaluating a new therapy. With regard to the exclusion of these clinical trials, two arguments were important to us: First, in clinical trials in which a new therapy is evaluated, the influence of this therapy on financial burden is investigated. However, this influence only refers to the evaluated therapy, so it would not really increase the external validity of the results. In addition, it is usually not yet known whether the evaluated therapies will be used in standard care. Second, clinical studies often use existing validated instruments to measure financial toxicity. The aim of our review was to explore which influencing factors on financial burdens generally exist and for which a significant influence has already been shown. The inclusion of clinical studies, in which the same PROM was used across most clinical studies, would not have contributed much to this question, as the same influencing factor would have been examined throughout. We fully agree with the reviewer that PROs reported under controlled conditions have a higher quality. Therefore, from our point of view, we would welcome that in further research the results of clinical studies in a new literature review compare the effects and effect sizes (measured under controlled circumstances) in order to be able to make a statement about which influencing variables have a stronger impact on financial toxicity. We have added a statement in the discussion/outlook to highlight this research gap. |
|
The consequence of such exclusion is that the quality of the selected papers is mostly fair and affected by a possible selection bias. Such a bias may be even more critical within small retrospective or cross-sectional studies. Sample size is a critical point for quantitative analysis considering that it was justified only in 2/30 studies according to what the Authors report. An exclusion based on some threshold of sample size could have been considered. Actually, the size of the included studies might be very small particularly for the qualitative ones (minimum 8 cases, median 22) but also for the quantitative ones (minimum 43, median 278); how many less than 100 patients? Some of the above criticisms might benefit of sensitivity analyses (eg. Including data from clinical trials, or excluding data from papers with less than 100 patients…). (R1) |
Thank you for your considerations. This review includes 4 quantitative studies with a sample size of less than 100 patients. Most factors in these studies did not show a significant effect on financial distress. The factors that did show a significant effect were age, health insurance and country. However, these factors did also show significant effects in studies with larger sample sizes. So an exclusion of studies with sample sizes smaller than 100 would not alter the results of this review.
We added a sentence in the results section to acknowledge this finding. |
|
The data of the section dedicated to quantified risk factors (para 3.3.3 and table 3) are weak due to quality limitations of the selected evidence papers. In the end, that something is reported as statistically significant in fair quality study probably affected by several selection biases does not mean too much for me. In addition, these data are confused and include statements that are internally inconsistent (as expected due to heterogeneity of the selected papers and acknowledged by the Authors). (R1) |
Thank you for pointing this out. We agree on that. With our review we aimed to map the currently existing knowledge about risk factors in third party payer systems illustrating the currently existing research gaps. The review highlighted that the research on that topic is only slowly evolving in universal healthcare coverage. Further research on that topic, especially in regards of valid instruments and study quality, is needed so that upcoming reviews rely on valid and consistent results and thus, increase external validity. |
Reviewer 2 Report
the title is too long. Words like systematic - underlying ... may be omitted and the overall title rephrased and shortened. The simple summary is redundant and different (obviously) from the abstract; unless it is required by the Journal standards, I would omit it completely.
The motivation of the study should be strengthened: why is it interesting? Which are the real literature gaps? You say something after Figure 1 but it is not enough.
You may well improve Figures 3 and 4 providing a better explanation that links the two.
Title of table 4 ... mistyping quantitative
make a check with grammarly.com or similar.
Expand a bit the conclusion with practical implications: why is Your paper useful?
Digiting Your title on Google Scholar
https://scholar.google.it/scholar?hl=it&as_sdt=0%2C5&q=Patient-reported+financial+burden+before%2C+during+and+after+can-cer%3A+a+systematic+literature+review+on+underlying+risk+factors+for+Financial+Toxicity+in+universal+healthcare+systems&btnG=
I find new unquoted references... some of them may be useful.
Author Response
|
Reviewer comments |
Response by authors |
|
the title is too long. Words like systematic - underlying ... may be omitted and the overall title rephrased and shortened. (R2) |
Thank you for your advice. We shortened the title to make it more precise. |
|
The simple summary is redundant and different (obviously) from the abstract; unless it is required by the Journal standards, I would omit it completely. (R2) |
The simple summary is part of the Journal standards, so we tried to differentiate between simple summary and abstract by focussing on the aim and the presented results, respectively. Otherwise, we would agree on your comment made. |
|
The motivation of the study should be strengthened: why is it interesting? Which are the real literature gaps? You say something after Figure 1 but it is not enough. (R2) |
Thank you for your recommendation. We restructured the introduction section after Figure 1 and expanded the argumentation about the importance of the topic. |
|
You may well improve Figures 3 and 4 providing a better explanation that links the two. (R2) |
Thank you for your hint. In our study we extracted the risk factors and its related occurrence based on the time points measured and questions ask within the paper. Once we classified the risk factors into suitable (sub-)categories, we transferred the occurrence domain onto an aggregated level of its underlying (sub-)categories. So Figure 3 and 4 are indirectly linked in terms of the classified (sub-)categories. We adjusted Figure 4 by marking the two domains that are linked to the main stages of the disease (diagnosis and survival) accounting for the different origin of the study population experiencing financial toxicity in the included papers. We also added a cross-reference in 3.3.2 and described the linkage of both paragraphs in a sub-clause. |
|
Title of table 4 ... mistyping quantitative (R2) |
Thanks for spotting, we corrected the typo. |
|
make a check with grammarly.com or similar. (R2) |
Yes, thank you, we double-checked spelling and grammar in the revision version by a native speaker. |
|
Expand a bit the conclusion with practical implications: why is Your paper useful? (R2) |
Thank you for the advice. We expand our conclusion illustrating the practical implications of our work for further research. |
|
Digiting Your title on Google Scholar, I find new unquoted references... some of them may be useful. (R2) |
Thank you, we really appreciate your effort made. We double-checked the references you found via Google Scholar and added some references within our introduction section and Riva et al (2019) in the discussion section. We tried to avoid referencing opinion papers due to its subjective bias, which effects some of the listed articles. The equivalent scientific papers (like Carrera or Zafar) are included within our paper. We also checked whether the other studies should be incorporated in our systematic review. Since most of them were conducted within improper country settings (especially US) or used financial toxicity as an independent variable, so that they did not match with our eligibility criteria. |
Reviewer 3 Report
I am very surprised that despite the large literature about the intolerable costs of anticancer drugs this is never mentioned in the paper.
Author Response
|
Reviewer comments |
Response by authors |
|
I am very surprised that despite the large literature about the intolerable costs of anticancer drugs this is never mentioned in the paper. (R3) |
Thank you for your hint, we also recognised the ongoing debate about the affordability of anticancer drugs in general. In our study, we focused on the individual financial consequences of a cancer disease rather than the systemic financial burden of healthcare systems due to cancer prevalence. As the included countries refer to a universal healthcare setting and are high income countries, it can be assumed, that even high anticancer drugs are reimbursed and co-payments are capped at a specific maximum. Therefore, the high costs of cancer drugs are not transferred to the individuals and do not mainly influence the degree of subjective financial distress (in comparison to the US setting e.g., where patients have to pay out-of-pocket for their medicines or partial co-payment in relation to the price of the drugs when having an insurance). We added a sub-clause within the introduction section to underline the additional strain for healthcare systems of subjective financial distress by mentioning the general trend about affordability of anticancer drugs. |
Round 2
Reviewer 3 Report
I think the article is suitabile for publication in Cancers
Author Response
Thank you very much